# Supporting Visual Comparison and Pattern Identification in Widescale Genomic Datasets

Venkat Bandi*          Carl Gutwin†

Department of Computer Science
University of Saskatchewan

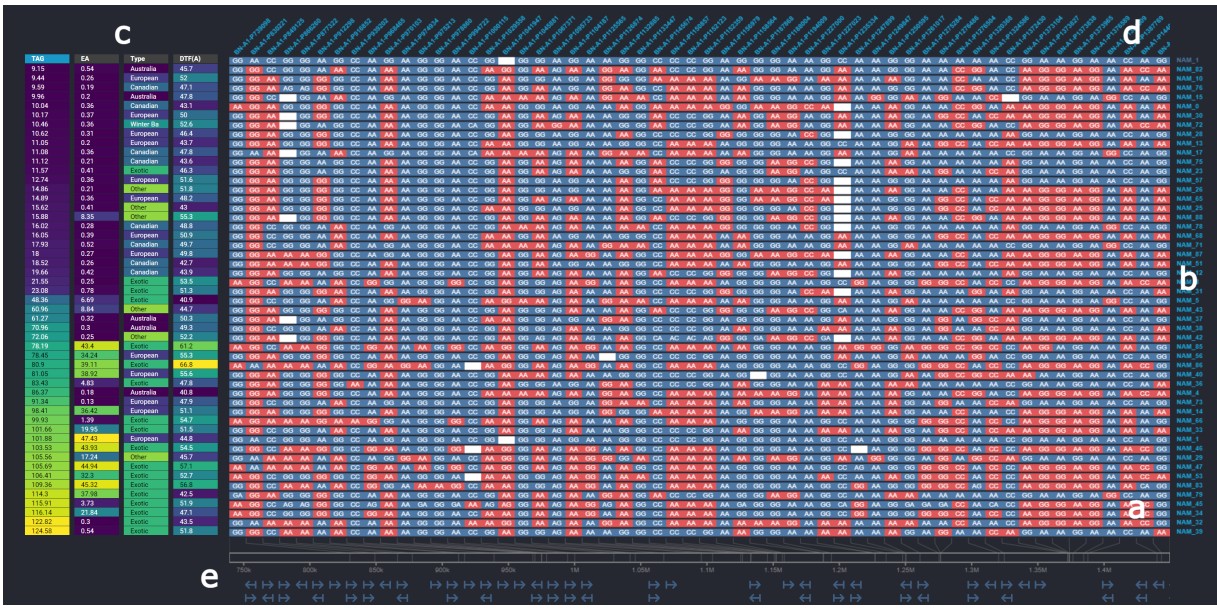

Figure 1: Visualization comparing SNPs at a specific genomic region in 52 varieties of Canola: a) Every SNP is coloured blue (match) or red (mismatch) based on its similarity to the reference variety at the top (first row), and missing SNPs are encoded in white. b) The name of each line variety is shown at right next to its corresponding row of SNPs. c) A reference map of phenotypic trait values is shown at left and the varieties are ordered by their increasing levels of aliphatic glucosinolates. d) The names of the SNPs are shown above each column. e) Connections to the SNP's genomic location are shown at bottom along with a map of genes in that region.

## ABSTRACT

Large-scale linear datasets are often visualized using tables. Visual analysis tasks in such systems involve comparisons and identification of patterns across rows and columns, but these tasks can be hard to perform as the table increases in size because rows and columns of interest can be far apart in the table. This problem is particularly evident in table visualizations of genomic data such as SNPs (which are genetic markers used in comparing different variants of an organism). Visual analysis of SNP datasets has a wide range of applications in plant breeding, genome-wide association studies, and pharmacogenetics. However, current SNP visualizations are limited in their support for complex analytic tasks in wide-scale tables. Through ongoing collaborations with genomic researchers and plant breeders, we have identified a set of new interaction requirements for visual analysis of SNP datasets, and we have developed a new visualization tool with interaction techniques that satisfy the requirements. Our requirements and techniques provide new understanding of how to support complex visual analysis in large-scale table visualizations.

*e-mail: venkat.bandi@usask.ca
†e-mail:carl.gutwin@usask.ca

**Index Terms:** Human-centered computing—Visualization—Visualization systems and tools—Visualization toolkits; Human-centered computing—Interaction design—Interaction design process and methods—User Interface design.

## 1 INTRODUCTION

In many visual-analytics domains, analysts use wide linear datasets that have many features or observations about a set of entities – for example, genomic data, time-series data, sequential documents, or population data. These datasets are often displayed using table visualizations in which each cell's value is encoded using a visual variable such as colour [69, 81, 101]. A main goal in working with table visualizations is to find insights from seeing patterns in the visualized data, such as determining that a particular row is different from a reference row in an important way, or that a particular column shows a pattern across the different rows, or that two columns show a similar (or contrasting) pattern to each other. These tasks involve two main activities in the visual workspace: finding patterns in the rows and columns that indicate potential relationships, and comparing rows or columns (either to a reference or to other parts of the data).

In the genomics domain, a common example of wide datasets is Single Nucleotide Polymorphism data. SNPs are genetic differences between genomes at a single base pair, and can be important in understanding the relationship of an organism's genotype to its phenotype (that is, its observable traits). For example, SNP analysis

is extremely common in plant breeding research, since SNPs have proven to be important markers for desirable crop traits such as flowering time, disease resistance, or protein content.

Plant breeders and genomic researchers are now able to quickly and easily produce datasets that collect large sets of SNPs (numbering from hundreds to tens of thousands) for many different varieties of a crop as shown in Figure 1, which is visualization of SNPs in 52 varieties of Canola. When a collection of SNPs are inherited together near a common loci they are referred to as a *haplotype* because they indicate a potential genetic linkage. Studying these clusters of SNPs and the DNA around them can help researchers identify specific mutations that affect the plant's characteristics, and can help breeders identify candidate genes for future crossings (although SNPs occur in both genes and in non-coding regions).

Many tools have been introduced that visualize SNP haplotypes, but few systems have focused on the interactive tasks that users need to carry out during exploratory investigations. Current tools are limited in their support for visual exploration – particularly in terms of lightweight visual comparisons in the wide datasets that are now common in breeding (often with thousands of columns). For example, mechanisms for navigating and comparing different columns in wide tables is of particular importance because most genetic locations in a plant's genome have dependencies that may be far away (for example, the polyploid nature of many plant genomes leads to multiple copies of genes on different chromosomes).

To better support visual exploration in wide datasets, we have been working with genomic researchers and plant breeders for several years to identify specific analysis tasks in large SNP tables, and interaction requirements that are needed to support those tasks. We identified the following six specific requirements:

- *Flexible and fast re-ordering mechanisms* so that users can quickly look at several arrangements of the SNP table, such as different domain-specific clustering and sorting methods as well as manual re-ordering;

- *Lightweight row comparisons* that allow temporary changes to encodings so that a quick comparison can be made without altering the overall organization of the table – for example, checking the difference between two rows without re-setting the reference row;

- *Comparisons between related columns* that allow multiple genetic locations to be compared even if they are far away in the table – for example, comparing SNPs at two locations that have orthologous genes;

- *Flexible encoding of differences* that allow users to rapidly switch between different visual representations of the difference between two plant varieties (for example, alternate colour schemes to show show existence of difference from a reference, 'cascading' differences, the type of difference, or the specific details for both varieties);

- *Support for location awareness* because the scale and organization of SNP table visualizations can lead to difficulty in tracking where a SNP is in the plant's genome (for example, whether a SNP is in an important region that is known to control other traits);

- *Managing and revisiting table configurations* to simplify navigation through the huge "configuration space" of ways that the user's current view of the table can be ordered, encoded, and positioned (for example, keeping track of what other clustering approaches have been tried, or how to get back to a previously-viewed configuration of the table).

We have developed a new SNP-haplotype viewer that provides novel interaction techniques to meet these requirements. The viewer provides lightweight mechanisms for arranging the table, comparing rows and columns, and looking at different encodings; it also shows explicit information about genomic and table location, and includes a configuration snapshot tool that provides automatic and manual saving of configurations as well as visualization of the saved snapshots so that they can be compared, revisited, and annotated.

Our work makes two main contributions: first, we identify several new interaction requirements for visual analysis of wide linear datasets – these arise from our collaborations in the plant-breeding domain, but there are several applications of the requirements to other types of wide tabular data; and second, we demonstrate new interaction techniques that can satisfy those requirements in a working genomics visualization tool. Our SNP-haplotype visualization is open-source and is freely available at *genomevis.usask.ca/haplotype-map* [3, 4].

## 2 BACKGROUND AND RELATED WORK

Three areas of prior work underlie our research: systems and techniques for table visualizations, techniques for and studies of visual comparison, and genomic visualizations of SNP data.

### 2.1 Visualizations of Tables

Tables have long been a standard way of communicating structured information using spatial layout. Table visualizations — which encode each cell's data value with a visual variable (colour, size, or position within the cell) -— have also been in use for more than a century, and have been well known since Bertin's work [10] and others as reviewed by Perin et al [81]). Table visualizations (sometimes called heat maps or colour-shaded matrices) allow large tables to be inspected and explored in a relatively small space, and tools for making visual tables are now a standard part of many visualization systems such as Tableau (*tableau.com*), PowerBI (*powerbi.microsoft.com*), and ggplot2 (*ggplot2.tidyverse.com*).

Table visualizations have been used in many different ways and in many different domains: for example, to summarize the characteristics of a set of locations [10, 45, 64]; to show the magnitude of a variable of interest (such as gene expression level or abundance of ions) for different samples [50, 70, 101]; to explore student engagement in online classes [21]; to explore database contents [58]; to show interactions in social networks [35]; to analyse energy demand over time for different buildings [105]; or to track employee performance through a set of criteria [106].

Some of the primary goals when visualizing tables are to help users understand relationships between the entities represented in the tables rows, the features or characteristics represented in columns, and associations between rows and columns. Analytics work in many domains where table visualizations are used is often equivocal and under-specified: for example, in the domain of genomics, Nusrat states "data visualization is essential for interpretation and hypothesis generation as well as a valuable aid in communicating discoveries. Visual tools bridge the gap between algorithmic approaches and the cognitive skills of investigators. [...] A key challenge in data-driven research is to discover unexpected patterns and to formulate hypotheses in an unbiased manner in vast amounts of genomic and other associated data" ( [76], p. 781).

Within this context, researchers have investigated many different aspects of designing, interpreting, and interacting with table visualizations. First, several projects have considered the problem of generating table visualizations: for example, Perin and colleagues revisited Bertin's early methodology for producing visual encodings inside table cells, and developed a tool for interactively creating table visualizations with a range of visual variables; others have developed tools for quickly creating table visualizations from spreadsheets [12] and arbitrary CSV files [14]. Researchers have also considered, how

to provide access to the table's values within the visualization: for example, Rao and Card's Table Lens provided a bar-chart encoding of cell values and mechansims for quickly sorting by column, and used a focus+context mechanism to allow detailed inspection of certain rows within the graphic presentation [85]; the Table Lens has also been extended by other researchers to allow multiple colour maps and clustering support [56]). A different approach was explored by Han and Nacenta, who created "Fat Fonts" that show both a scalar value and provide a visual representation of the value through amount of ink [41]. Table representations have also been adapted to show hierarchical data [28, 60].

Second, many researchers have investigated ways of ordering and arranging a table to best reveal patterns in the data. Careful manual arrangement of rows and columns was an important part of Bertin's original methodology [10], and many tools allow manual reordering of rows and columns. However, with larger datasets, manual ordering is not feasible, so automated algorithms for clustering or "pattern mining" [26, 54] are often employed – these can use similarity to create a tree from the table's rows [101], or can look for visual patterns in the table data [11, 24, 56, 59, 81].

Third, many systems provide explicit support for specific tasks, such as ranking candidates [38, 99], interactively looking for patterns [11], navigating through versions of tables that change over time [82], extracting and comparing data subsets from different tables [36], interaction techniques for working with event sequences [40], or dimensionality reduction [11, 28].

Finally, matrix visualizations are a subtype of table visualizations in which the two dimensions of the table represent the same features for two entities, and each cell represents a degree of association between the entities for that feature. Matrix visualizations have also been used in many domains: for example, to show graphs and networks [9, 47, 48], term co-occurrence [35], genomic similarity [42], physical connections in folded structures [22, 102], software evolution [89], or classification errors like confusion matrices [34]. Researchers have also investigated several novel representations for matrices, including dual views that pair a matrix with its corresponding node-link diagram [46], integration of matrices into existing node-link structures [48], extensions that allow display of multivariate data [104], and 'matrices of heatmaps' to increase the number of dimensions that can be shown [88].

## 2.2 Supporting Visual Comparisons in Visualizations

Comparisons are a common and frequent task in visual analytics, and techniques for supporting comparison have been widely studied. Many techniques can be classified using the three approaches proposed by Gleicher: juxtaposition, superimposition, and explicit encoding [30, 31, 66]. *Juxtaposition* involves placing visualizations in close proximity, in order to allow users to see similarities and differences in parallel parts of the visualizations. For example, if two line charts are presented side by side, viewers can compare values and trends in the charts (as long as all representations use the same layout and scale so that visual differences accurately reflect differences in the underlying data). A common technique that juxtaposes several visualizations is the small-multiples method [10]: each of the multiples has a similar layout but different data, allowing comparisons by looking across the images. This idea has been used in many ways, including well-known techniques such as scatterplot matrices [49], as well as extensions to immersive environments [61]. Juxtaposition can also be achieved interactively: for example, Tominski's CompaRing approach brings comparison candidates close to the cursor when the user selects an object [95].

*Superimposition* involves putting two datasets in the same visualization so that differences are visible in the same reference frame. For example, instead of showing two line charts side by side, the two lines can be shown in the same chart. Using a common reference frame allows similarities and differences to be seen more clearly – however, this method has the problem of clutter, and the density of some representations mean that they do not work well as overlays (for example, space-filling methods or dense data spaces), and the approach works best with sparse data (although the visual presentations can be adjusted to reduce occlusion).

*Explicit encoding of a comparison* involves creating and visualizing a new dataset that explicitly represents a specific comparison -– e.g., the data from two line charts can be used to create a new dataset showing the difference between the lines, and then this new dataset can be shown explicitly as a new line (either in addition to or instead of the existing lines). Many types of explicit encoding are possible: for example, showing the existence of differences, the magnitude of differences, or the type of differences (limited only by the ways in which two datasets can be compared) [73]. Researchers have demonstrated several explicit-encoding methods in visualization research, including colour-based differences (such as showing same/different colouring, or amount of difference), "diff matrices" that show a matrix of line pairs [92], annotations that indicate differences in one of the representations being compared (for example, coloured lines showing missing or added elements in a tree [13]), differences between tables at different time periods [73], changes between video frames [15], or "shine-through" representations to highlight differences in overlays [96].

Researchers have also extended Gleicher's three basic categories to include other representations. Different visualizations can be presented sequentially in the same location, either using the idea of Rapid Serial Visual Presentation (RSVP) [7], or using animation to smoothly morph from one dataset to another [25]. This technique is a combination of juxtaposition and superimposition using time (temporal juxtaposition), and can address the occlusion problem while still making use of the common spatial frame. Tominski showed a variation on this idea in a technique that allowed the user to 'peel back' a top representation to look at the bottom representation [96]. Other researchers have extended the idea of juxtaposition by nesting one visualization inside another, which allows different types of comparisons [53], and have introduced the concept of overloading one representation with details from another, such as showing graph elements that are present in one visualization but not in another [52].

In addition to comparison approaches based on spatial layout, researchers have also considered the actions and interactions that are part of visual comparison tasks. For example, von Landesberger specified the workflow involved in a visual comparison task [98]; Wu developed a "view composition algebra" to understand and compose actions in ad-hoc comparison settings [103]; Jardine and colleagues investigated the low-level perceptual processes involved in visual comparison [51]; and Kehrer and colleagues defined a formal model of category comparisons in small-multiple displays [57]. An additional higher-level consideration is the amount of effort required to carry out a visual comparison – low-effort techniques are critically important for supporting effective exploration of large datasets. A few researchers have explicitly focused on effort reduction – for example, Tominski's CompaRing which reduced the steps required to juxtapose two comparators [95].

Studies have also been conducted to look at the performance of different techniques for supporting visual comparisons. Early perceptual studies investigated performance on comparisons between elements in bar charts [16] and individual differences in same/different visual comparison tasks [17]. Several studies have followed up on these results to look at comparisons in standard chart types [93, 94] the effect of chart size and space usage on interpretation [43], and the effect of glyph types on reading and comparing time-series visualizations [27]. Several researchers have evaluated the basic processes involved in visual comparison: for example, Lu and colleagues created a model of just-noticeable differences as the basis of visual comparison, and explored this idea with bar charts, bubble charts, and pie charts [65], and Ondov and colleagues studied low-level

perceptual tasks to compare performance in several presentation styles (overlays, small multiples, and animated transitions) [79]. Other studies have considered specific representations or analysis scenarios: for example, user performance in visual comparison, slope estimation, and discrimination tasks for multiple time-series visualizations [53]; the performance of square and triangular matrix representations as well as different methods of matrix juxtaposition [63]; the effectiveness of small multiples compared to animated transitions for seeing changes in graphs [2]; and user performance when comparing ranked data in tables [8].

## 2.3 Genomic Visualizations and SNP Haplotypes

There are many types of genomic visualization that are used to show a wide range of information – for example, sequences and sequence alignment, levels of gene expression or ion abundance, conserved regions of the genome (synteny), or structural variation across different samples [1, 5, 20, 67, 68, 76, 86, 107] – see [76] for a broad survey. In particular, recent advances in sequencing capabilities and the increasing availability of genomic data has led to the use of genetic analysis and genomic visualization in the domain of plant breeding where one of the main goals is to connect a crop plant's *genotype* to its *phenotype* – the observable characteristics or traits of the plant. Plant breeders and genomic scientists investigate how genetics affect important crop traits such as oil and protein content, plant height, resistance to disease, or heat tolerance; this knowledge can be used to create hypotheses and choose candidates for breeding in order to try and introduce and retain desirable traits [55].

Although complete sequencing of individual genomes is still time-consuming, it has become feasible to identify large numbers of genetic markers in a genome using the "genotyping-by-sequencing" approach [19, 83] that generates sets of markers called SNPs for a variety. SNP markers are often associated with differences in traits of interest, and so SNP visualizations are an important part of marker-assisted breeding [55].

Several systems have been developed for showing SNP data, including capabilities in general-purpose genomic visualization tools (for example, JBrowse [20] or Gosling [67]) as well as dedicated applications such as Haploview [6], Flapjack [69], SNP-Vista [91], or GCViT [100]. These systems often show table visualizations with individuals in rows and SNPs in columns, as well as association matrices that show co-occurrence of different alleles within a haplotype group [6], or histograms of SNP counts within a given window size [100]. Many tools provide clustering capabilities, such as using a genetic-similarity dendrogram [91]) as well as interactive zoom to let users see details of the alleles and the actual nucleotides. A few tools are paired with algorithms for conducting genome-wide association studies (GWAS) that look for correlations between SNPs and measured traits of interest [33]. However, there are still many limitations in current genomic visualizations in terms of support for the task of interactive visual comparison, although a few examples of research that focuses on comparison do exist: very early work developed diagrammatic methods for comparing DNA sequences [29]; Glueck and colleagues developed the PhenoBlocks visualization with the goal of supporting comparisons across phenotypes [32]; Mitra and colleagues developed methods for comparing metagenomic datasets [71]; and recent research by Ripken and colleagues conducted requirements interviews with biologists for working with genomic data in a VR environment for immersive analytics – the identified requirements included the need to compare data subsets, and the need to flexibly reorder and group the data [87]).

A specific limitation of current SNP-haplotype viewers is that most tools have been primarily built for analysis of diploid genomes (common in humans or animals) whereas plants are often polyploid, with multiple copies of each gene [62]; breeders and researchers often need to consider the effects of all orthologous locations together during exploration, but simultaneous visual access to orthologues is not well supported in most tools. The drawbacks of current tools and our collaborations with plant breeders and genomic researchers led us to the new requirements and visual features described below.

## 3 APPLICATION DOMAIN

To contextualize the design of a visualization tool for SNPs, we provide an overview of the biological background for the domain, and a characterization of the dataset used in the visualization.

### 3.1 Biological Background

Genomics research involves the study of an organism's DNA in order to understand its structure, function, and evolution [80, 84]. An organism's complete set of DNA is called its genome, consisting of a large set of nucleotides that encode the instructions responsible for the organism's development and function [72]. There are four nucleotide bases – Adenine (A), Guanine (G), Cytosine (C) and Thymine (T). A variation in a single nucleotide in the genome at a specific position is called a Single Nucleotide Polymorphism or SNP. These variations tend to exist in a significant fraction of the population (1% or more) and the different variants of a particular SNP are called *alleles*. When a set of SNPs that are adjacent to each other in the genome are inherited together they are referred to as a haplotype. Mapping the location of these haplotypes can help researchers in classifying different variant populations.

### 3.2 Data Characterization

SNP data can be represented in different types of files such as VCF (Variant Call Format) or Hapmap (Haplotype Map) and is often analyzed in combination with additional data sources such as a GFF (General Feature Format) file for the position of genes, and a phenotypic-trait table. At the most basic level, however, SNP data is ordered based on genomic position and classified according to the population line (variety) such that each SNP has the following features:

- Identifier: Every SNP is given a unique identifier that is common across all the different parental lines of a single organism.

- Possible Alleles: The different nucleotide variants that exist for a SNP; while most common SNPs have two alleles, triallelic SNPs have been identified in human genomes.

- Position: The location of a SNP in the genome, typically encoded relative to a chromosome.

- Value: The nucleotide variant present in the given population line; the value can be empty when the data is missing.

Table visualizations of SNPs use the inherent ordering, and then build a table at the genome, chromosome, or region level. Other datasets can supplement the SNP information to indicate, for example, the gene that the SNP is on, or copy number variations at that genomic location. Further, other data sources can provide additional information about each variety: phenotypic traits such as flowering time, protein content, or seed size; or dendogram trees that cluster the lines based on their genetic distance. These additional datasets are primarily used to control the order of the rows.

## 4 REQUIREMENTS FOR SNP-HAPLOTYPE ANALYSIS

We have been working with genomic researchers and plant breeders over the past five years to understand user tasks and requirements for visual exploration in genomic datasets. Our collaborating research groups are interested both in producing new crop variants that have improved agronomic or nutrition traits, and also in exploring genetic evidence for hypotheses about physiological mechanisms and plant evolution. One research group is interested in using our

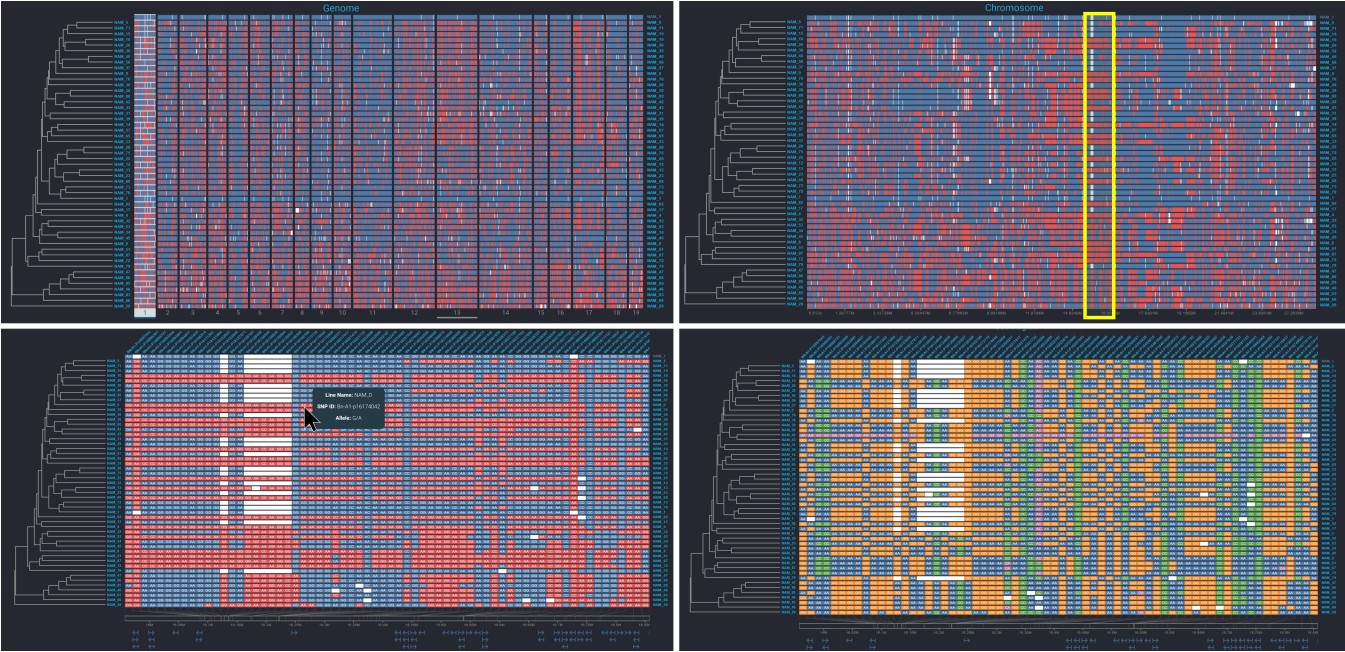

Figure 2: The SNP browser's three main views: genome-level overview (top left) showing all 19 chromosomes of Canola, with chromosome 1 selected; chromosome-level view with highlighted viewfinder rectangle (top right); region view with match/difference colouring (bottom left) and hover popup showing details; region view with nucleotide-based colour scheme (bottom right).

visualization system to map the differences among multiple Lentil germplasm/stock lines. They are particularly interested in combining SNP data with CNVs (copy number variations) to break down the lentil population into smaller clusters and identify candidate genes [39]. Another group wants to use our system to investigate the genetic diversity between founder lines in a Canola breeding population [23]. A third research group that investigates Blackleg (an oilseed pathogen fungus) is interested in analyzing SNPs across the common Blackleg isolates relative to the position of avirulence genes. Requirements analysis has been carried out in an iterative and collaborative fashion with these research groups, and we have developed and deployed several versions of our haplotype visualization – the prototypes have been used as a foundation for discussions about user tasks and visual-exploration needs. Based on our discussions, we have identified the following requirements that go beyond what is available in current SNP visualizations:

$R_1$. **Flexible and fast re-ordering mechanisms**. Genomic crop analysis involves looking for associations between SNPs, genes, and traits of interest – and to do this, users need to be able to quickly look at several arrangements of the SNP table. For example, ordering rows by genetic similarity, sorting by a measured trait, clustering by allele group for a particular SNP, or arranging rows manually (based on the user's knowledge of the varieties) are all common manipulation methods for our collaborators. In addition, it is valuable to be able to move between these different arrangements quickly and easily.

$R_2$. **Lightweight row comparisons**. Because there are many ways in which varieties can be compared, users need lightweight mechanisms for quickly seeing how one row compares to another without changing the global ordering of the table. In addition to simple selection of a reference variety that changes the global visualization, there is a need for low-effort ways of comparing any two given rows. For example, in a table that is colour-coded based on differences from a single reference,

users need a way to do a quick comparison of the differences between two varieties without changing the overall reference.

$R_3$. **Comparisons between related columns**. A genetic location in a plant genome is often related to other locations: for example, many plant species are polyploid (multiple duplicated copies of genes across the genome), and many genes also have dependencies with other parts of the genome (such as when a gene in one location may be regulated by another). This means that users need to compare the columns of a table visualization as well as the rows – and need easy access to related locations, since a SNP table may be many thousands of columns wide.

$R_4$. **Flexible encoding of differences**. There are many ways in which genomic researchers think about the difference between two varieties: they may be interested simply in the existence of differences between a variety and a reference; they may want to see specific differences at the nucleotide level; they may be interested in exact matches between alleles or partial matches (heterozygous nucleotide pairs); or they may want to see 'cascading' differences that build up across multiple varieties. Alternate encodings, for example using colour maps, can show different kinds of differences, but users need to be able to switch between encodings quickly and easily.

$R_5$. **Support for location awareness**. The size of SNP table visualizations (often in the order of tens of thousands of columns) means that it can be difficult for users to maintain awareness of where they are in the genome – a problem that is exacerbated by the fact that SNPs are simply ordered in the table, rather than positioned relative to their actual genomic location. As a result, it is critical that any visualization provide support for awareness of location, both at a high level ("what chromosome am I looking at?") and at a low level ("what gene is this SNP on, and how many neighboring SNPs are on the same gene?").

$R_6$. **Managing and revisiting table configurations**. With multiple ordering mechanisms, multiple colour encodings, and zoom

and pan navigation, there are an enormous number of possible configurations for the table visualization. It can be very difficult for users to remember where they have been in this "configuration space" and how they can get back to a previous configuration (for example, to show a pattern to a colleague or to revisit a previous candidate). Although provenance tools have been introduced for several visualization systems [18, 37], no current genomic visualization systems (to our knowledge) provide any support for this requirement.

## 5 SYSTEM OVERVIEW

Our haplotype browser is a web-based application for visualizing and exploring SNP groups across multiple varieties (parental lines) of crop species such as Canola (*Brassica napus*), lentil (*Lens culinaris*), or wheat (*Triticum aestivum*). The system provides several table visualizations at different genomic scales, with varieties in the table's rows and SNPs in the columns (see Figure 2). After the user selects or loads a datafile, the system displays a genome-wide overview of all varieties and SNPs, divided into chromosomes. Since there are often many thousands of SNPs for each variety ( 30,000 in the Canola dataset of Figure 2), this table is highly compressed horizontally, and so primarily serves as a consistent frame of reference that helps the user orient themselves to the data and keep track of navigational cues such as the zoom region. The main user interaction at the overview level is to select a chromosome for closer analysis, which is then displayed as a second table below the overview.

The chromosome view uses the same tabular organization as the overview, but at a higher zoom level, where users can start to identify patterns in the data and locations for closer investigation — for example, the central region of the chromosome view in Figure 2 shows that there are a number of varieties that differ in terms of several contiguous SNPs. To zoom in further on this region, the chromosome view provides a viewfinder rectangle that selects a subset for a third view that shows only the region of interest (yellow rectangle in Figure 2).

The region view is shown at the bottom of Figure 2. When the zoom level is high enough in this view, the names of the SNPs are shown at the top of the table, and the actual base pairs are also drawn in the table cell. In this view, several additional interactions are available. The user can pan (by dragging) and adjust the zoom level (using a slider above the view), and can hover over any cell to show a tooltip with information about the SNP and its corresponding alleles. Button toggles are provided above this view for the user to move left or right across the region in small step increments to investigate neighbouring SNP clusters. There also a pair of input boxes to enter a specific start and end position if the user is targeting a known genetic loci. All three views use the same basic encoding scheme, as described in the following section.

## 6 VISUAL ENCODING DESIGN

SNP data is primarily visualized through a simple coloured tabular grid where the level of detail changes depending on the genomic resolution. In encoding this dataset we followed previous SNP genotype visualizers like FlapJack and Haploview [6, 69, 74] that plot the parental lines horizontally with colored SNP markers running vertically. We extend this design space in our visualization by providing three panels: a main SNP panel and two supporting panels of associated data, with coordinated interaction support among all three for complex analysis tasks. The main panel, visualizing the SNP markers, is at the center of our visualization. To its left is the line ordering panel that encodes the ordering of the parental lines either via a dendogram tree or a heatmap of phenotypic traits. The final panel is positioned underneath the main panel and visualizes the genetic-to-physical location map of the SNPs and the corresponding genes around the loci. The visual encoding of all three panel is flexible and can change based on a variety of interaction and selection parameters.

### 6.1 Main SNP Panel

The main table visualization has several possible colour encodings – some of these are based on comparisons of each line to a reference line (shown at the top of the table), and some based on underlying genetic information. The multiple color encoding schemes are designed to meet requirement $R_4$.

The first (and default) color scheme is an explicit encoding of differences to the reference line: if a SNP allele in a particular line matches with the SNP allele in the reference line, it is painted blue and, if there is a mismatch, it is painted red. Since each allele is inherited from one parent, the alleles are always shown in pairs and can be homozygous (same) or heterozygous (different alleles in the pair). Since most SNPs have two possible alleles ( for example A/C), the three possible genotypes could be either a homozygous pair of the first allele (AA) or a homozygous pair of the second allele (CC) or a heterozygous pair of both (AC or CA). In the default color scheme, a SNP is considered to match if one among the pair of alleles is the same as the reference (and is thus painted blue). The second color scheme is a variation of the first, and ignores partially-matching SNPs such that a marker is painted blue only if the alleles from both parents match the alleles in the reference SNP.

The third color scheme is used to investigate the homozygosity of SNP clusters – it paints a SNP marker blue if the pair of alleles within the SNP are the same, or red if they are different. This can help researchers isolate parental lines with a higher concentration of heterozygous SNP pairs. The fourth color scheme uses the underlying DNA, with the SNP marker colored based on the nucleotide bases present in the alleles. There are four basic colors used for each of the four homozygous base pairs (AA, GG, CC and TT) and all heterozygous base pairs are painted purple. This colour scheme is shown in Figure 2 (bottom right), where many SNPs show two main groups with either the AA or the GG allele. A fifth and final color scheme is used to visualize similarity among lines in a cascaded fashion with each line colored based on its similarity with all the lines above it (see Section 7.3 below). In all five color schemes, missing data where a SNP is not present in a line or its allele is unknown is painted white.

The organisation of the table visualization is based on the genomic resolution. At the whole-genome level, the SNPs are grouped into chromosomes in order to provide an overview of the dataset and also highlight large-scale patterns. For example, a large clusters of missing SNPs either across the lines vertically or in a single line horizontally can indicate an error during sequencing or the SNP assaying process. It also provides spatial context for the user as they investigate SNP clusters in a specific region. When a chromosome has been selected, it is highlighted using a white background in the genome view. Canvas rendering at this level is optimized through an algorithm that filters out minuscule SNP variations to improve rendering speed. This optimization occurs automatically when the size of the rendered SNP markers goes below a single pixel.

In the chromosome view, painting of the SNP markers is the same as at the genome level, but with the addition of a viewfinder window that allows selection of a region for closer analysis. In the region view, SNP markers are painted using the chosen colour scheme, along with a label in each cell indicating the pair of alleles in the SNP. At this resolution additional markers can also be painted on top of the SNPs such as copy number variations. These are either insertions or deletions in genes at specific locations across the genome and are highlighted as red or white circles with white circles indicating insertions and red circles deletions as shown in Figure 3.

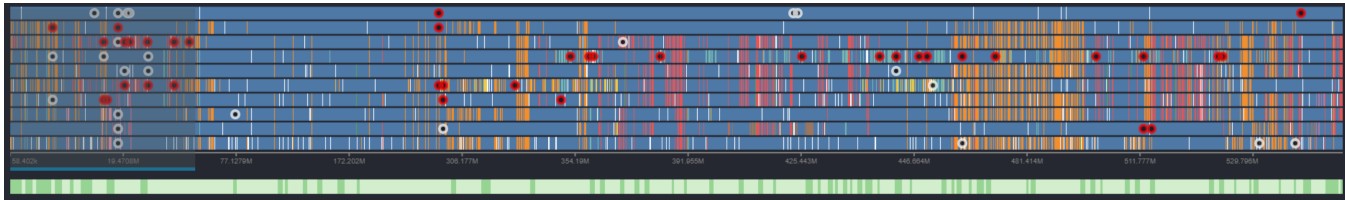

Figure 3: Visualization of copy number variations in the chromosome view (red circles indicate deletions and white circles indicate insertions). The green ribbon underneath the SNP panel is visualizing gene density at the SNP loci with dark green locations indicating a higher density.

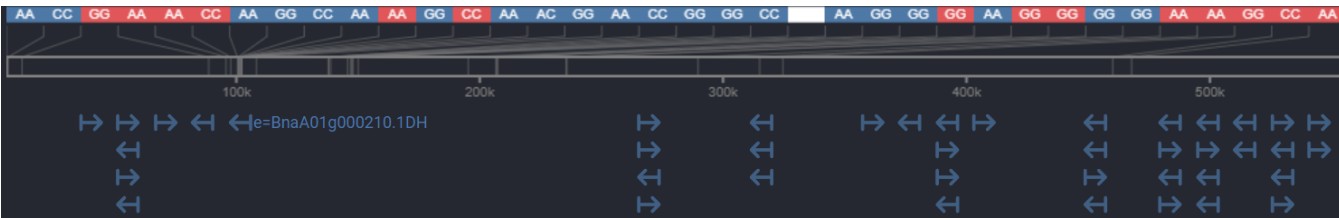

Figure 4: Visualization of genes as pointed arrowheads indicate their position and orientation in the genome. The fine gray lines are connecting SNPs with their physical location in the genome.

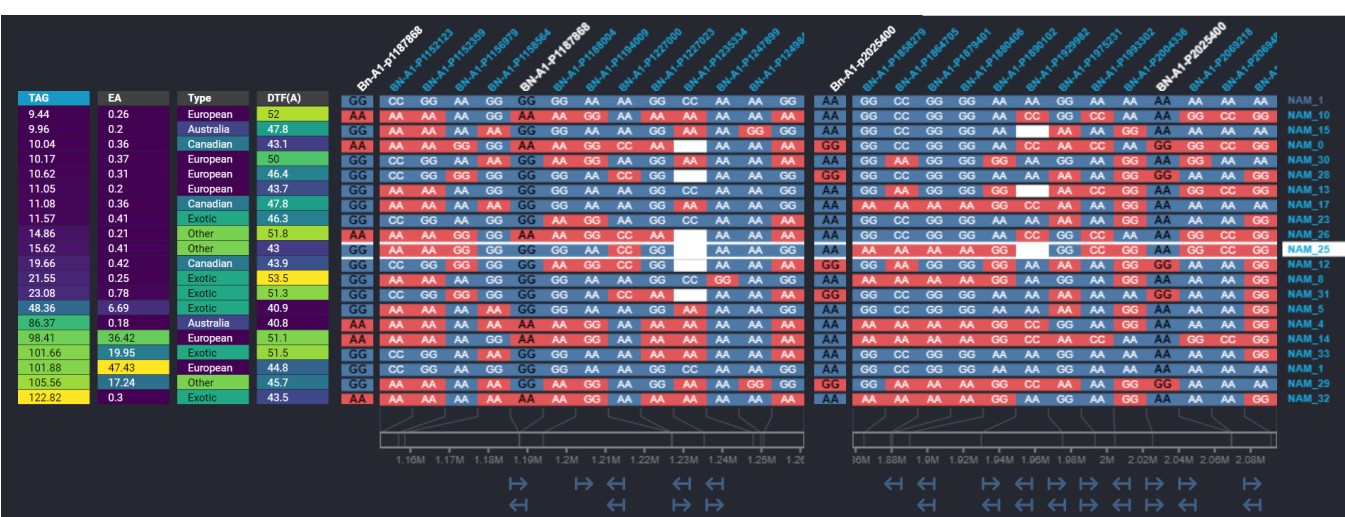

Figure 5: Split View demonstrating comparison of SNP clusters across two different genomic regions in the same chromosome. Each view has a SNP column pinned to the left and the line NAM¨25 is selected and highlighted across both views.

## 6.2 Line Ordering Panel

This panel is designed to meet requirement $R_1$. The ordering of the different parental lines is important to researchers because several insights can be gained by identifying similar regions in the table's columns – this is because the extent of similarity in the SNP clusters around a loci across the lines is an indication of shared ancestry or origin between the lines. By default our visualization system orders the lines based on a dendogram tree provided by the user. This tree structure visualizes every parental line as a leaf node in a tree, and it clusters lines based on evolutionary distance. This arrangement can help researchers in studying the SNPs of a particular subset of the lines that are similar to each other.

The other ordering mechanism consists of heatmaps of different phenotypic traits for each of the parental lines. The trait map contains one column for each trait (seed size or protein content), with colouring based on a heatmap of the range of values for that trait. The *Virdis* color palette is used for the heatmaps for easier distinction between the lines [75]. The lines can be ordered by sorting them based on any of the column values, which places lines with similar phenotypes closer to each other. This feature is explored further in the interaction design section below.

## 6.3 Gene Loci Panel

SNPs in the main view are ordered from left to right based on genetic position. However, because SNPs may be unevenly distributed across the genome, the position of a SNP's column does not match its physical location in the genome. This makes it difficult to visually indicate additional information regarding the genetic loci of the SNPs. To address this problem and meet requirement $R_5$, we provide a visual map that provides the entire genomic scale of investigation underneath the SNP view and connects every SNP to its actual physical location in the genome, as shown in Figure 4. Additional datasets sujch as gene density maps or gene markers are then placed underneath this physical map so that they corresponding to the location of the SNPs. For the whole-genome view, this panel is hidden as the density of lines makes it difficult to discern positional information. In the chromosome view, the panel is used to show a simple scale indicating the actual physical location of the SNPs in terms of number of base pairs, and can be used to highlight additional datasets such as gene density tracks. In the region view, the panel shows individual genes located near the loci of the SNPs. The genes are visualized as horizontal arrows, with the direction of the arrow indicating the orientation of the gene (see Figure 4). Clicking on a gene arrow shows the gene ID and additional information related to the function of the gene or the protein it encodes.

## 7 INTERACTION FEATURE DESIGN

Here we outline the different interactive design features in our visualization that address the six major requirements.

## 7.1 Dynamic Ordering of Lines

Users are given several option to order the different parental/variety lines. By default, lines are ordered according to a dendogram tree based on an input file provided by the user. This mechanism clusters lines that are evolutionarily similar. If a dendogram file is not available, the lines are sorted automatically based on the SNP similarity with the reference line. This approach ensures that matching SNP clusters get pushed to the top of the main view while the lines that differ the most are pushed towards the bottom. Additionally users are also given the option of manually selecting a subset of lines through a multi select dropdown list. The order of lines in this case is determined based on the order in which the lines are selected. This gives researchers the option to investigate specific patterns that they might have observed in the dataset in greater detail by only comparing those lines.

If a file containing phenotype trait values is provided by the user, then the lines can also be ordered based on these traits. Users are first given an option to select the traits they are interested in mapping for all available traits in the file (the order of selection determines the placing of the trait columns from left to right). Then users are given an option to order the lines based on a specific trait value. This ordering can be changed by clicking the column head of any phenotype trait in the trait map. This offers a fast and flexible way to reorder the lines thus meeting requirement $R_1$.

## 7.2 Navigating Multiple Genomic Resolutions

When investigating large-scale datasets, users need to be able to navigate quickly while still maintaining contextual information regarding their position in the dataset. We provide location context through the three coordinated views described above (genome, chromosome, region). Navigating from genome to chromosome involves clicking on the desired chromosome, and then selecting a region involves positioning the viewfinder window. The viewfinder is translucent by default to ensure that it does not occlude the view of the chromosome, and has a darker border at the bottom indicating the region that has been selected (Figure 3). The user can drag the viewfinder and adjust its left and right extents with the mouse.

In scenarios where SNP density is high in a chromosome, it might be difficult to use the viewfinder to zoom into a small enough region due to the limited size of the window. To address this issue, a navigation panel is available in the region view to aid users in controlling the region of interest. It contains two input boxes to enter genomic start and end position (base pair locations) from the start of the chromosome. This allows researchers to look at all SNPs near a specific gene loci (for example, a gene that corresponds to a particular protein). The view also included navigation buttons that let the user move the region in small incremental steps, and a slider provides additional control over the zoom level of the region view. As the user interacts with the navigation panel, the corresponding changes are reflected in the viewfinder in the chromosome view, maintaining location awareness between the views and from the table to the genome.

## 7.3 Dynamic Color Scheme

Apart from the four basic color schemes discussed above, we also offer users a novel way to compare a small subset of lines through a cascading waterfall color pattern. This feature was designed to meet requirement $R_4$. When users manually select fewer than ten varieties for comparison, the visualization changes into a dynamic color scheme for visualizing accumulating differences, instead of the standard blue and red scheme. In the cascade color scheme, every line is compared with all the lines above it instead of just the one reference line at the top. To encode the similarity pattern, every line is first assigned a unique color. Then all the SNPs in that line are compared with the lines vertically above it starting from the top. If a SNP markers matches with any of the lines above it, the color of the topmost matching line is assigned to the SNP. If the marker doesn't match any of the lines above it, it is considered novel and is painted in the unique color assigned to the line. This ensure a cascading waterfall style of coloring, such that all the SNPs in the first line have the same color because there are no SNPs above them. In the second line, all the SNPs that match the first line are painted in the color of the first line and all the SNPs that do not match are painted in the color of the second line. This flow continues until the final line is a mixture of different colors of all the lines above it depending on the precedence of SNPs markers present in it. This offers researchers a insight into the origin of a unique cluster of SNPs. However this color scheme only works for ten lines or fewer, due to the limitations on the number of colours that users can reliably distinguish.

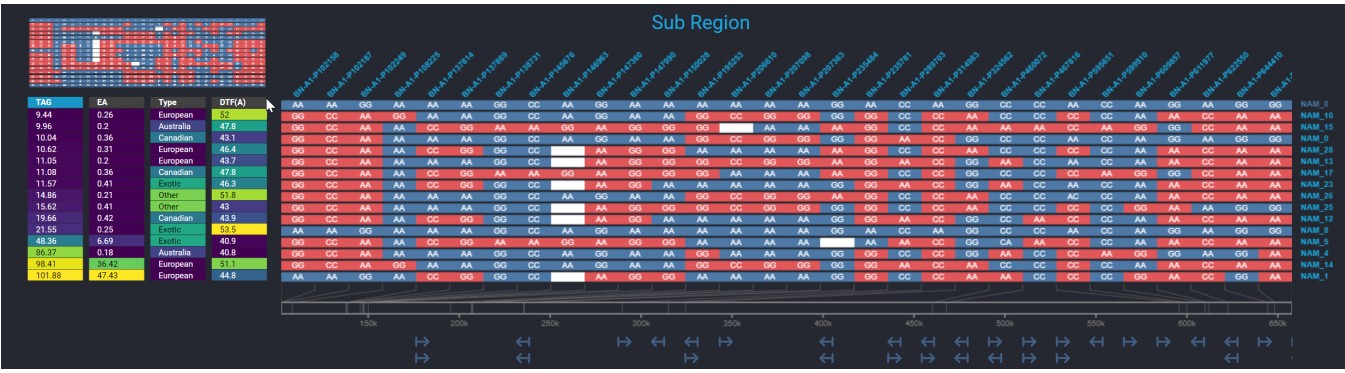

Figure 6: A preview of the SNP view is shown in the top left corner above the phenotype trait map when the user hovers their mouse over a particular phenotype. The preview shows what the SNP view would look like if the lines were sorted using that particular phenotype.

## 7.4 Row and Column Highlighting

In large datasets with many rows and columns, it can be hard for users to navigate across the SNP view to identify a specific SNP marker and its allele annotations. To support users with this task, we offer a row highlighting option that lets users highlight a specific row by clicking on the line name (this draws white guidelines across the SNP view as shown in Figure 5). To further aid users with this problem, we also offer a tooltip feature that shows all the details such as the corresponding line name and the SNP index of a specific marker when the mouse is hovered over it. The guidelines and tooltips greatly improve the user's ability to trace SNPs along a row – and the highlighting can also act as a temporary landmark that allows visual inspection of rows above and below the guidelines.

Another issue that can occur during visual comparison of two SNP columns is the distance between their loci. If the SNPs are far apart, they cannot be viewed in the region window without zooming out and losing detail. To solve this problem and meet requirement $R_3$, we introduced a column-pinning mechanism (inspired by spreadsheet applications) that lets users pin a SNP column to the beginning of the region view. The selected SNP column is also highlighted in the region view by changing the color of the allele annotations on each marker in the column from white to black. Users can then pan across the chromosome to a different location and compare the SNP columns in that region with the pinned SNP column (see Figure 5, in which two SNP columns have been pinned in the region view). An additional advantage of this feature is that it also lets users temporarily mark and highlight a SNP column that might have caught their interest for further investigation.

## 7.5 Multi-Region Analysis

Although SNPs are inherited in clusters around a specific gene loci, several SNP clusters across the genome can be related to each other due to gene duplication or dependency. This is a common issue for polyploid plants which may have several duplicated copies of the same gene. The regulation and expression of these genes can vary based on the SNP clusters within or around them, which means that researchers often have to jump between multiple regions in the genome to compare these SNP clusters. While the column-pinning feature discussed above helps in this situation to an extent, it only lets users compare a single column at a time, meaning they lose context of the neighbouring SNPs in the cluster. To address this problem, we implemented a split-screen view that splits the region view into two parts with each part focusing on a different region. All of the other features discussed above such as row and column highlighting get carried over into these split views. This gives users the option to pin two different SNP columns and compare their neighbouring clusters in a side-by-side view (as shown in Figure 5).

This feature allows our system to meet both requirements $R_3$ and $R_5$.

## 7.6 Lightweight Comparison Preview

Based on the feedback collected from our collaborators, one of the most commonly used selection features is the ability to switch the reference row at the top and compare a different line with the other lines. In certain datasets such as lentil (*Lens culinaris*) the number of lines that are being studied are quite high due to the large number of possible cultivars and variants. This is a general problem in most food crops as they are cultivated across the world in a variety of environmental conditions with different outcomes due to the selective breeding process. This means breeders often need a way to carry out lightweight row comparisons across the lines 'on the fly' without switching the reference line of the entire visualization. To address this issue and meet requirement $R_2$, we offer a preview mode along with the row highlighting feature. Users can first select a row in the SNP view by clicking its line name. This highlights the entire row with white guide lines. Users can then hover their mouse over any other row to perform a quick comparison between the two rows and update the coloring in the row that is being hovered. The coloring switches back to its default state once the mouse hover over the row is removed. This way researchers can quickly compare select rows without having to switch the main reference row at the top and update the whole SNP view. Similarly, we also offer previews for the interactions in the trait map that re-orders the table: users can hover their mouse over the column head of a trait column to see a quick preview in a small floating window next to the mouse cursor of what the SNP view would look like if the lines were ordered based on that phenotype. The preview disappears as soon the mouse is moved away from the column head.

## 7.7 Revisitation Support Through Snapshots

The exploratory nature of our visualization tool means that users will interact with the dataset at multiple resolutions and in complex filtering scenarios – and it can become difficult to switch contexts between different viewpoints for visual comparison when looking at SNP markers in different regions. To help address this issue and meet requirement $R_6$, our system maintains an in-memory store of the sequence of actions that led to the current state of the visualization. Each of these memory states are stored along with a thumbnail image (snapshot) of the visualization at that instant. A floating snapshot panel that is minimized by default is available for users to pull up and explore prior states of the visualization. Users can then click on any of these snapshots to go back to the state of the visualization at that prior point in time. This providers users with a lightweight history tracking mechanism that can help them retrace their steps during data exploration. The snapshots are automatically tagged

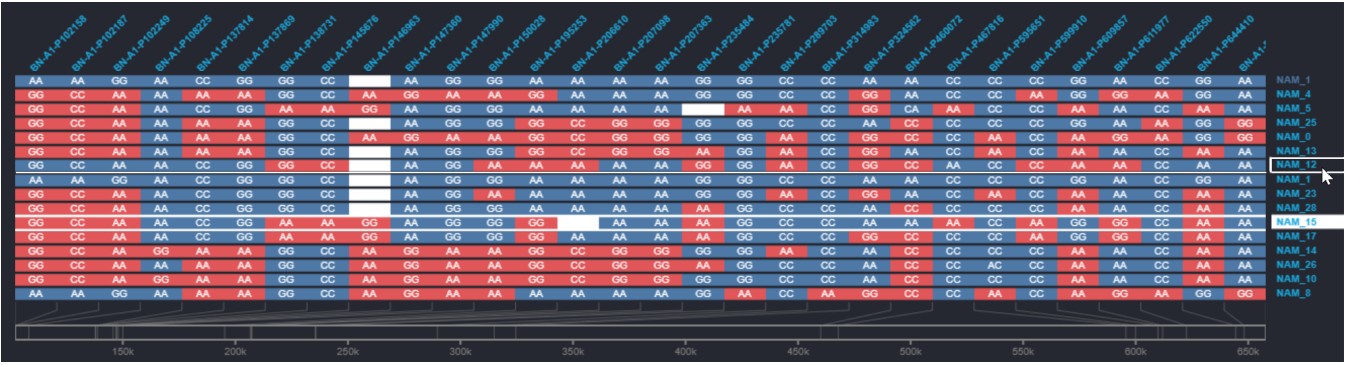

Figure 7: A preview of the light weight row comparison feature. When users hover their mouse over a row after highlighting a different row. The hovered row is highlighted with a single guide line and its colouring is update to reflect similarity with the other highlighted row instead of the reference line at the top.

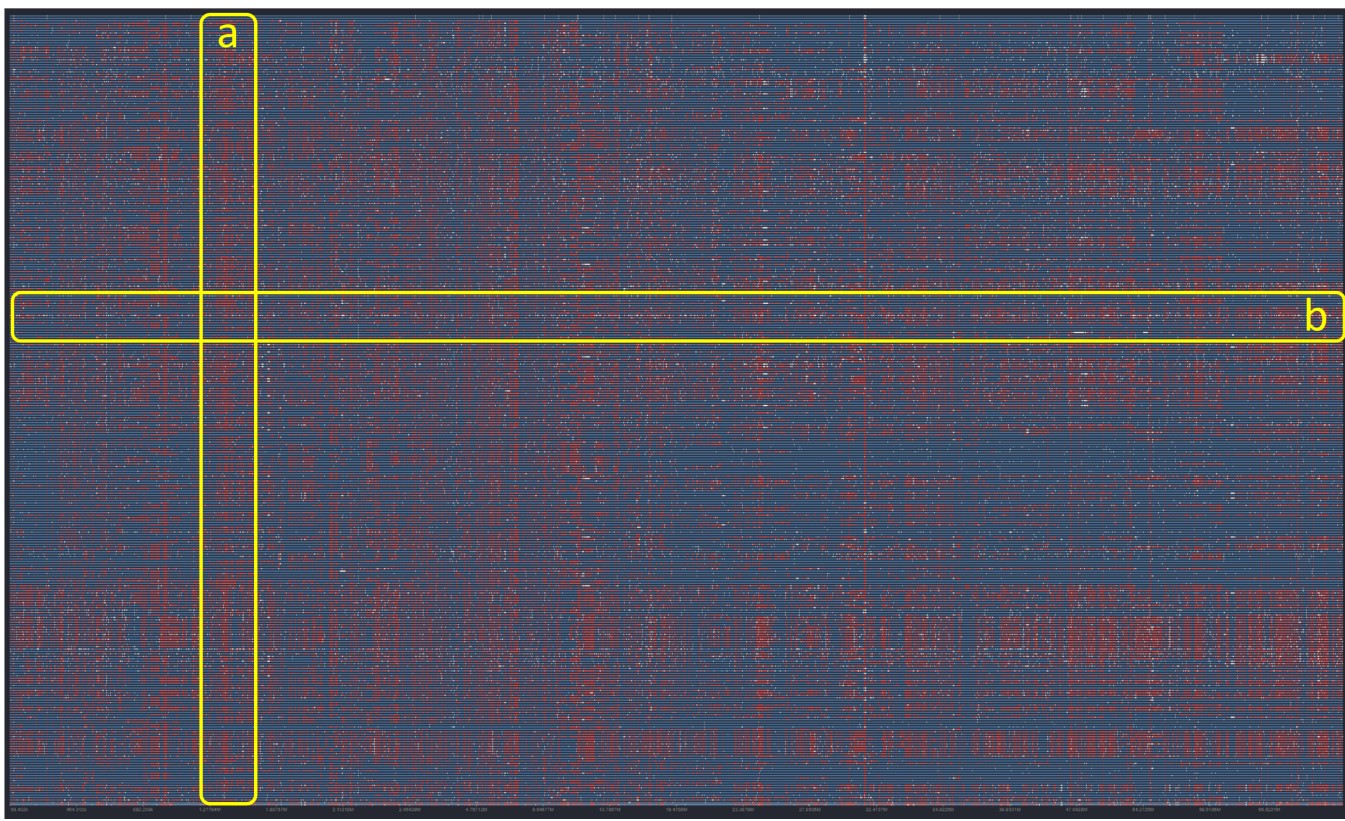

Figure 8: SNP View visualizing the similarities between 10K SNPs in a reference line across 328 Lentils varieties. a) Cluster of SNPs that dont match with the reference line across most of the varieties. b) One lentil variety line seen as an almost white line across the entire region indicating missing data across its entirety due to possible sequencing error.

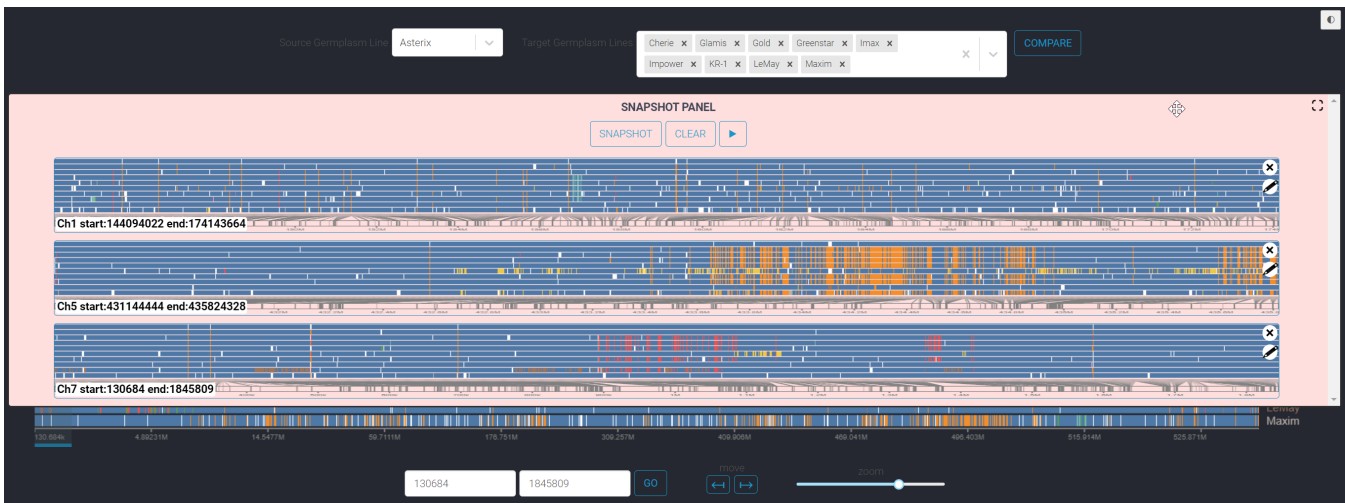

Figure 9: A example of the snapshot feature that lets users store their interaction history as a series of snapshots with thumbnails showing the state of the visualization when the snapshot was taken. The panel is minimized by default but can opened up as seen here and contains three snapshots.

with a note that indicates the chromosome name and the start and end position of the region view. This note can be edited by users to also include other points of interest if needed. The snapshot feature also provides users a novel way of interacting with the system by creating snapshots of multiple regions of interest and going back and forth between them for quick visual lookup and comparison. The system includes mechanisms for automatic creation of snapshots (if the user stays in a particular configuration for 30 seconds) as well as manual creation (through a keyboard shortcut).

## 8 ITERATIVE REFINEMENT, TESTING, AND CURRENT USE

The design of our visualization system was iteratively refined over a period of two years through multiple rounds of feedback from our research collaborators as they used our system to explore their different datasets. During this period our system was also stress tested with larger datasets ( 29,000 SNP markers across 1000 lines of barley, and the first 10,000 SNP markers across 328 lentil varieties developed at the Crop Development Center in Canada).

An example visualization of the large lentil dataset is shown in Figure 8; this demonstrates the usability of our system even with very large tables – even at this scale, the visualization shows genome-level patterns such as lines with an extensive set of missing markers (8 (b)) or SNP clusters that are completely different across the majority of lines (8 (a)). Our system is also in use by a group of plant breeders to showcase the diversity of agronomically important traits among a population of Canola founder lines, and has been adapted for use with several other use cases including exploration of genotypes for Blackleg, a common oilseed pathogen. Our tool is open-source and freely available [3, 4], and has been integrated into KnowPulse, a major North American pulse crop database to visualize the differences among their various lentil and other pulse cultivars [77, 90]. Our web tool has also been integrated into an accelerated breeding portal developed at the Global Institute of Food Security in Saskatchewan [78].

## 9 DISCUSSION AND FUTURE WORK

In the following sections we consider the relationship of our requirements and techniques to previous work on table visualization and visual comparison, discuss ways that our techniques can be applied to datasets outside the domain of genomics, and outline a set of directions for future work.

### 9.1 Requirements and Techniques in Context of Previous Work

Working in real-world collaboration with genomic researchers and plant breeders means that our SNP-haplotype viewer implements some interaction techniques that are shared with what has been seen in previous systems – for example, two of our requirements match those identified in Ripken's interviews with biologists [87], although Ripken's research took a broader view and our requirements are thus more focused on the comparison tasks themselves; similarly, several systems have provided techniques for clustering, sorting, and manual row rearrangement [50, 56, 70, 101, 106], and column pinning and split-screen views are common in spreadsheet applications (although not seen in tabular visualization systems).

However, several techniques and features are novel (or have novel adaptations to fit the scenario of large-scale SNP tables). First, our techniques for row and column comparisons are an advance in terms of user effort: the lightweight row comparisons, column pinning, split-screen views, and visual previews of column sorting substantially reduce the number of steps needed to carry out a visual comparison. Reducing effort in exploratory visual analysis is critical: although there may be ways to achieve the comparison using standard techniques, it is important to provide low-effort mechanisms so that users can follow exploratory paths without needing to think about multiple steps in the UI. Our goals here are similar to those of Tominski's CompaRing system [95], although his approach used juxtaposition whereas ours uses an explicit encoding of difference. Second, techniques such as providing three persistent zoom views that follow the structure of the genome, and visual tracks to indicate genomic location as well as gene commonality, assist the user in maintaining location awareness (since table locations are not well matched to actual locations). Third, providing snapshots to track, compare, share, and revisit table configurations is an extension to the work done previously on visualization provenance [37, 44] that broadens the focus from communication and storytelling to supporting the basic mechanics and processes of navigating through the complex parameter space of table configuration.

### 9.2 Generalizing to Other Types of Wide Datasets

Although we have focused on the domain of genomic research and the specific needs of our collaborators, we believe that several of our requirements and interaction techniques will be applicable to

other domains as well. Column-based comparison tools will be useful whenever the data has columnar dependencies or links between columns. For example, if columns are used for temporal data, there may be cyclic relationships that need to be brought closer together for investigation (natural cycles such as seasons, or links created by external phenomena such as temperature data during sunspot years). Flexible row comparison mechanisms will also be important in any dataset where there are many entities, and where comparisons need to be made between rows as well as to an obvious reference row. For example, a dataset of baseball players (as was used in the Table Lens [85]) does not have a single clear reference, and it is likely that many different pairs of players could be compared for a given task. The idea of multiple flexible encodings can also be useful in other datasets – these allow users to cycle quickly through different perspectives on the comparison, gaining a broader view of differences. In particular, our 'cascading differences' encoding could be useful in showing the accumulation of changes when rows represent successive versions of a complex entity, such as a software code base. Finally, a configuration-snapshot mechanism should also be widely applicable in any visualization where users change organization frequently, and where users need to revisit recent configurations that they have previously explored.

### 9.3 Future Research Directions

Our future work will involve activities to improve the SNP viewer as part of our ongoing collaboration, and new projects to explore broader visualization issues raised by our experience. In the SNP visualization system, we will add algorithms for pattern mining in the table data [11, 24, 59, 81] and tools for comparing these patterns to external evidence such as GWAS results. We will also add support for additional context tracks (for example, to provide GWAS results or other gene-centric measurements such as expression level) – aligning GWAS results such as a Manhattan plot with the table visualization can provide a bridge between algorithmic approaches and visual analysis [76], and gives users a set of starting points for their exploration. We also plan to extend the interactions available with the configuration-snapshot tool, such as to provide explicit encoding of differences between two configurations [13, 73, 97].

In the broader visualization context, our initial goal is to test our new interaction techniques in other types of wide tabular datasets, and broaden our interaction requirements to encompass new tasks and comparison activities: we will work with datasets that use continuous rather than discrete values (requiring new encodings for the table), we will test our tools with large-scale time series that contain cyclic column dependencies, and we will add additional techniques to work with table subsets [36, 87]. We also plan to follow up on work that has looked at the details of visual comparisons [51, 57, 103] and assess the components of visual comparison in table visualizations (and support for these components) at a more fine-grained level.

### 10 Conclusion

Analytics tasks in large-scale table visualizations involve comparisons and identification of patterns across rows and columns, but these tasks become more difficult when tables are large – as is the case for SNP analyses in genomic research. Current SNP visualizations are limited in their support for complex analytic tasks in wide-scale tables – both because they do not focus on interaction, and because they do not address issues raised by tables with thousands or tens of thousands of columns. In collaboration with genomic researchers and plant breeders, we have identified six new interaction requirements that will help to support visual analytics tasks with wide-scale SNP datasets. The requirements cover needs for flexible arrangements of the table, lightweight comparisons of both rows and columns, flexible visual encodings, and the ability to save table configurations. We developed a new SNP-haplotype

viewer that implements interaction techniques for each of our proposed requirements; the tool has been in continuous and successful use by our collaborators over several years. Our work contributes both a better understanding of the needs for large-scale visual analysis in table visualizations, and specific interaction techniques that can address those needs.

**ACKNOWLEDGMENTS**

The authors wish to thank Isobel Parkin, Andrew Sharpe, Kirstin Bett, Lacey-Anne Sanderson, and Larissa Ramsay for the expert knowledge they brought to this project. This work was supported in part by the Natural Sciences and Engineering Research Council of Canada (NSERC), and additionally by the Plant Phenotyping and Imaging Research Cenre.

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
