# OpenReview forum: "Supporting Visual Comparison and Pattern Identification in Widescale Genomic Datasets"
_graphicsinterface.org/Graphics_Interface/2023/Conference_SD — GI 2023 - second deadline_

### Official Review · Reviewer_sRQX · 2023-04-13
**lack of novelty and no evaluation**

**Rating:** 3
**Confidence:** 5

**Review:**

This paper presents a visualization tool that support visual comparison of gemonic datasets formatted in large tables. The paper is well written. The related work seems thorough, however, lacks the clarity about the differences of the proposed tool and existing systems. There exist two big concerns that prevent the paper from publishing at GI23.

The first is the novelty. Like reviewed in the literature survey, there exist tons of table/matrix visualization systems already. It is unclear what the unique advantages of the proposed tool, compared to existing ones. For example, [1] also visualize very large matrix and supports a hierarchy on the side. I don’t see a clear novelty of this work.

[1] A. Bilal, A. Jourabloo, M. Ye, X. Liu and L. Ren, "Do Convolutional Neural Networks Learn Class Hierarchy?," in IEEE Transactions on Visualization and Computer Graphics, vol. 24, no. 1, pp. 152-162, Jan. 2018, doi: 10.1109/TVCG.2017.2744683.

Secondly, there is no evaluation of the effectiveness and/or usefulness of the proposed visualization tool. Sec 8 only briefly mentions that the tool is iteratively refined over two years and used by their collaborators. However, there lack sufficient details regarding this iterative process. How was the system refined? What were the early versions look like? What did the collaborators say regarding the pros and cons for each iteration? Additionally, a comparative user study is usually expected to verify the design of the tool. Without a rigorous and solid evaluation, it is hard to assess if the system is useful.

Overall, I think this paper is still in its early stage and requires future work on evaluation to be publishable.

---

### Official Review · Reviewer_4BWP · 2023-04-23
**Well-written and passes the research smell test - but extremely hard to evaluate without more domain expertise.**

**Rating:** 7
**Confidence:** 2

**Review:**

This paper submission presents a specialized matrix visualization tool tailored to support the examination of Single Nucleotide Polymorphism data. The proposed design and features seem to be compelling extensions of existing matrix visualization techniques, and several (including views supporting comparisons between related columns and supporting location awareness) seem quite likely to be useful for other kinds of large-scale matrix data. Broadly, the paper does a good job of surveying related work on matrix visualization and interaction, and the introduction, description of the design, and discussion are all polished and read well.

However, as a reviewer with essentially no biology experience (let alone a background in crop genomics) I was unable to meaningfully follow the discussion of SNPs and it's unclear to me *WHAT* the visualizations are showing or whether they do so in an effective manner. As the paper focuses primarily on describing the approach, rather than making claims about its effectiveness, I feel more confident still recommending acceptance than I might otherwise. However, this lack of clarity and accessibility to the GI audience (and my inability to assess these aspects of the contribution) are a meaningful caveat to my review. The paper *feels* compelling and polished, and it seems to address a set of problems that could plausibly be impactful for geneticists working in this area. I see relatively little harm in publishing it at GI, but suspect that the main audience for this work probably lives in a different community, and that the novel visualization contributions would probably be better demonstrated with more accessible data.

With those caveats in place, I have just a few suggestions for improving the publication:

* **Annotate the figures!** I found it quite difficult to parse any of the visualizations shown in the paper. While this almost certainly due in part to my lack of experience in this domain, the figures also lack any sort of meaningful pointers, annotations, and other context to help readers read and interpret them. The captions often cryptically refer to elements in the figures in ways that are difficult to follow. Annotating all of these images to highlight and explain the important pieces of the interfaces, the novel features they're intended to illustrate, and (probably most importantly) *what* those features enable users to see could greatly improve the accessibility of the paper.

* **Provide more detail about collaborators and collaboration!** The paper is frustratingly vague about the collaboration with genomics researchers and plant breeders. Other than the fact that the collaboration has lasted for ~5 years, I can tell basically nothing about what kinds of researchers were involved, how many, and in what contexts. More information here (even if anonymized) could make the paper's claims feel more credible.

* **Share more evidence of the system in action!** The paper suggests that the tool is open-sourced and being used, but obfuscates all links to the system and provides no video or other evidence of the tool in use. Being able to play with the system or see explanatory video of others doing so could make it much easier for readers to appreciate how it works and what it's able to do. The handful of cryptic images in the paper really don't do it justice.

* **Lay off the e.g. and i.e.** While it is generally well-written, the paper uses the Latin abbreviations "e.g." and "i.e." close to 80 (!!!) times — almost always in ways that degrade the quality and readability of the text. In essentially every place this abbreviation appears in the paper, a plain-English equivalent ("including","like","for example", "such as") would lead to much clearer and more readable text. In many cases (including in most parentheticals and references), these abbreviations can simply be removed. Every place you use them, imagine a reader encountering the full Latin phrase ("exempli gratia" or "id est") and you might get a better sense of just how disruptive the terms can be. As used here, they often feel like a sloppy writing crutch that mar an otherwise legible document. I'd strongly encourage trying to remove them from the text entirely.

---

### Official Review · Reviewer_gPQf · 2023-04-23
**The paper describes a system for analyzing and comparing genomic datasets using an intuitive table metaphor and associated interactions. Overall, the paper is well written, albeit verbose. The interactions described are not particularly novel and the authors do not report any formal evaluation of the tool beyond stating that researchers have been using their tool successfully.**

**Rating:** 7
**Confidence:** 4

**Review:**

The authors present a tool for analysis of genomic data, specifically, Single Nucleotide Polymorphism (SNP) datasets, using an intuitive table-based visualization tool. The authors also describe the design and implementation of interactions that are specifically beneficial for the analysis of this type of genomic data. The authors also enumerate six requirements for SNP analysis that are used to guide the design of their approach. The authors report on the validity of their approach by stress testing with extremely large datasets and anecdotal accounts of researchers using the tool successfully. Overall, the problem is well motivated and the authors are able to describe a need for this approach in the field of genomics.

However, the interactions described are not particularly novel - pinning columns, split screen, row and column highlighting etc. are well-established interaction techniques for comparison and analysis. SNP-haplotype datasets are shown to be very complex, leading to very wide table, but that is not specific to SNP datasets. That said, the techniques that are developed and reported are still shown to be useful for their specific use cases.

The authors do not report any results from an evaluation in a formal design study. They report that breeders and researchers have used this system successfully and that it is integrated in a major North American pulse database. This makes it difficult to assess the validity of their approach. A qualitative analysis would have greatly strengthened the paper.

Overall, the authors make a good contribution to the field of table based visualizations and provide a detailed analysis for the requirements of analyzing and visualizing complex genomic data. The authors can strengthen the work by providing clearer description of why their interactions are novel and also providing the results from a formal evaluation study, which may not be possible in the given time.